MGST2 and WNT2 are candidate genes for comitant strabismus susceptibility in Japanese patients

Zhang Jingjing
Matsuo Toshihiko matsuot@cc.okayama-u.ac.jp
Department of Ophthalmology, Okayama University Graduate School of Medicine, Dentistry, and Pharmaceutical Sciences , Okayama City , Okayama , Japan
Buyske Steven
Electronic publication date: 2017 Oct 17
Publication date: 2017
Volume: 5
Electronic Location ID: e3935
Received 2017 Jun 2; Accepted 2017 Sep 26
Copyright: ©2017 Zhang and Matsuo
Copyright year: 2017
Copyright holder: Zhang and Matsuo
License: This is an open access article distributed under the terms of the Creative Commons Attribution License, which permits unrestricted use, distribution, reproduction and adaptation in any medium and for any purpose provided that it is properly attributed. For attribution, the original author(s), title, publication source (PeerJ) and either DOI or URL of the article must be cited.
License URL: https://creativecommons.org/licenses/by/4.0/

Keywords: Comitant strabismus, Linkage analysis, Exotropia, Esotropia, Case-control association study, Transmission disequilibrium test allowing for errors (TDTae), Chromosomal susceptibility locus, Japanese families, Transmission disequilibrium test (TDT), Candidate gene

Funding: Japan Society for the Promotion of Science 23659811 This study was supported in part by a grant-in-aid (23659811, Toshihiko Matsuo) for Challenging Exploratory Research from Japan Society for the Promotion of Science. There was no additional external funding received for this study. The funders had no role in study design, data collection and analysis, decision to publish, or preparation of the manuscript.

==============================
Background/Aim

Strabismus is a common condition with misalignment between two eyes that may lead to decrease of visual acuity, lack of binocularity, and diplopia. It is caused by heterogeneous environmental and genetic risk factors. Our previous research has identified new chromosomal susceptibility loci in 4q28.3 and 7q31.2 regions for comitant strabismus in Japanese families. We conducted a verification study by linkage analysis to narrow the chromosomal loci down to a single gene.

Methods

From Japanese and U.S. databases, 24 rsSNPs and 233 rsSNPs were chosen from the 4q28.3 and 7q31.2 region, respectively, and were typed in 108 affected subjects and 96 unaffected subjects of 58 families with primary and non-syndromic comitant strabismus. Three major analytical methods were used: transmission disequilibrium test (TDT), TDT allowing for errors (TDTae), and linkage analysis under dominant and recessive inheritance.

Results

The SNPs with significant P values in TDT and TDTae were located solely at the gene, microsomal glutathione S-transferase 2 (MGST2), on chromosome 4q28.3 locus. In contrast, significant SNPs were dispersed in a few genes, containing wingless-type MMTV integration site family member 2 (WNT2), on chromosome 7q31.2 locus. The distribution of significant SNPs on the 7q31.2 locus showed that only the ST7 to WNT2 region in the same big haplotype block contained significant SNPs for all three methods of linkage analysis.

Conclusions

This study suggests that MGST2 and WNT2 are potential candidates for comitant strabismus in Japanese population.

Introduction

Strabismus refers to misalignment between two eyes that point in different directions, and is classified into comitant (concomitant) strabismus and incomitant (noncomitant or paralytic) strabismus. Structural anomalies of extraocular muscles, such as anomalous insertion, hypoplasia and aplasia, have long been recognized as congenital causes for hereditary incomitant strabismus (Matsuo et al., 1988; Uchiyama et al., 2010; Okano et al., 1990; Matsuo et al., 2009a). More recently, genomic mutations or polymorphisms have been identified in families with hereditary incomitant strabismus, including congenital fibrosis of extraocular muscles (CFEOM), Duane syndrome (Engle, 2007; Graeber, Hunter & Engle, 2013), or congenital (idiopathic) superior oblique muscle palsy (Jiang et al., 2004; Jiang et al., 2005; Imai et al., 2008; Ohkubo et al., 2012). Congenital progressive external ophthalmoplegia is also well recognized as a mitochondrial disease with hereditary background. Acquired incomitant strabismus is caused by vascular, traumatic or compression paralysis of ocular motility cranial nerves. It may also be developed as a consequence of muscle diseases, or resulted from hyper- or hypothyroidism, myasthenia gravis and other rare conditions.

Primary and non-syndromic comitant strabismus is a multifactorial disorder which has both genetic and environmental background with their undefined contribution (Michaelides & Moore, 2004; Maconachie, Gottlob & McLean, 2013; Ye et al., 2014). Genetic influence is evidenced by family history (Abrahamsson, Magnusson & Sjostrand, 1999; Matsuo, Yamane & Ohtsuki, 2001; Taira et al., 2003) and phenotypic concordance between monozygotic twins (Podgor, Remaley & Chew, 1996; Matsuo et al., 2002; Sanfilippo et al., 2012). Environmental influence is supported by the association with premature birth and perinatal hypoxia as it occurs with a higher incidence in cerebral palsy (Cotter et al., 2011; Jacobson & Dutton, 2000). At present, no gene has been identified to be responsible for the development of comitant strabismus. American and British researchers reported 7p22.1 as a chromosomal susceptibility locus for esotropia in Caucasian families (Parikh et al., 2003; Rice et al., 2009). Our previous research has identified the susceptibility loci in 4q28.3 and 7q31.2 regions for comitant strabismus that comprised both esotropia and exotropia in Japanese families (Fujiwara et al., 2003; Shaaban et al., 2009a; Shaaban et al., 2009b). Other chromosomal loci have also been reported to be associated with comitant strabismus in other ethnicity (Khan et al., 2011; Bosten et al., 2014).

Given this background, we conducted single nucleotide polymorphism (SNP) analyses to narrow the chromosomal loci down to a single gene in Japanese families. As an analytical method, we previously tried to use association study that examines the relationship between several polymorphic markers and the strabismus phenotype in the chromosomal regions (Matsuo, 2015). In this study, we used three different methods for linkage analysis: transmission disequilibrium test (TDT) (Spielman & Ewens, 1996), TDT allowing for errors (TDTae) (Gordon et al., 2004), and linkage analysis under dominant and recessive inheritance (Lathrop et al., 1984). We hypothesized that the results by different analytical methods of linkage analysis might localize a specific gene that would be responsible for comitant strabismus.

Materials and Methods

Subjects

This study involved 108 affected subjects and 96 unaffected subjects in 58 Japanese families with primary and non-syndromic comitant strabismus including both esotropia and exotropia, which mostly overlapped with subjects in the previous study for chromosomal loci identification (Shaaban et al., 2009a). The previous study used 55 families with at least four members in each family. Part of the genomic DNA samples that were used in the previous study were no longer available due to the DNA shortage. Thus, the present study involved new affected subjects and unaffected subjects in new families as well as available subjects of the previous study. The features of 58 families are summarized in Table 1. The study followed the tenets of the Declaration of Helsinki, and was approved by the Ethics Committee of Okayama University Graduate School of Medicine, Dentistry, and Pharmaceutical Sciences (Approval No. Genome 215).

Table 1 Clinical features of affected subjects with primary and non-syndromic strabismus and unaffected subjects in families.

	Affected (%)	Unaffected (%)	
The number of individuals	108 (52.9%)	96 (47.1%)	
Male	45 (53.6%)	39 (46.4%)	
Female	63 (52.5%)	57 (47.5%)	
The number of families	55 (94.8%)	3 (5.2%)	
Exotropia	22		
Esotropia	25		
Mixed phenotypes (exotropia and esotropia)	8		
The number of individuals	108 (52.9%)	96 (47.1%)	
Exotropia	52		
Intermittent exotropia	44		
Constant exotropia	8		
Esotropia	56		
Infantile esotropia	33		
Accommodative or partially accommodative esotropia	14		
Microtropia (Microesotropia)	3		
Unclassified strabismus	6		

SNP selection and typing

Tag SNPs in the two chromosomal regions were first picked from the JSNP database for Japanese (Hirakawa et al., 2002). Finally, 24 rsSNPs were chosen at the 4q28.3 region and 233 rsSNPs were chosen at the 7q31.2 region from the dbSNP database of the US National Center for Biotechnology Information (NCBI). Genomic DNA that was isolated from peripheral blood leukocytes was amplified by multiplex polymerase chain reaction (PCR).

The MassARRAY system is a high-throughput matrix-assisted laser desorption ionization and time-of-flight mass spectrometry (MALDI-TOF MS) for detection of nucleic acids. After PCR-based multiplex reaction and clean-up to remove unincorporated dNTPs, the third primers were introduced into the reactions which correspond to DNA template immediately at front of polymorphic sites. Single nucleotide base extension reaction was performed with mass-modified nucleotides. Then SNPs were identified on the platform of MassARRAY Analyzer 4 (96 well) iPlex SNP Genotyping (Sequenom, San Diego, CA, USA). Overall call rates were 87%. We then proceeded to quality controls of SNPs and samples.

Hardy-Weinberg equilibrium and principal component analysis

We first performed Hardy-Weinberg equilibrium (HWE) testing for data quality control. Principal component analysis (PCA) was performed by the genome-wide complex trait analysis (GCTA) program (Yang et al., 2011) to calculate eigenvectors which were then put in the model as covariates to identify if there be a population substructure among families.

Family-based association study: TDT and TDTae

Transmission disequilibrium test (TDT) is a test for association in the presence of linkage for a case-parent trio. Thus, two parents had to be present in a pedigree with one affected subject. As it is customary to only show paternal (or maternal) in a pedigree drawing, we assigned all information unknown except for their sex in genetic analysis, but we did not expect parent-specific effects so that we treated the maternal and paternal genotypes symmetrically. Both parents with homozygous condition were not informative, as there was no genetic variation at the locus in the progeny. Only subjects with at least one heterozygous parent were informative. This situation led to reduction of effective sample size.

Plink program version 1.9 (Purcell et al., 2007; Purcell & Chang, 2015) was run to detect genotypes which violated the Mendelian rules in TDT analysis. The Plink program can handle errors by rendering the offending genotypes “unknown”. Therefore, it was run on the original data without deletion of any SNPs or families which went through the initial cleaning. TDT is a form of linkage analysis which is only powerful in the presence of genetic association (Ott, 1989). The errors were handled by Plink to eliminate the offending data in TDT analysis as mentioned above.

In contrast, transmission disequilibrium test allowing for errors (TDTae) is an implementation of TDT which allows errors to be present in estimating their rates in the course of analysis by TDTae program (Gordon et al., 2004). In the process of running the TDTae program, errors were estimated in the background of any one of a number of error models. We considered the two most reasonable and economical error models, which require few parameters: DSB (Douglas-Skol-Boehnke) allows for genotype errors (Douglas, Skol & Boehnke, 2002) and GHLO (Gordon-Heath-Liu-Ott) allows for allele errors (Gordon et al., 2001; Yang et al., 2008). DSB is analogous to error models proposed many years ago and has only been implemented in the last decade. Under the two error models, we run TDTae program for dominant (d), recessive (r) and multiplicative (m) inheritance.

Furthermore, we defined linkage disequilibrium (LD) blocks, using Haploview 4.2 (Gabriel et al., 2002), on chromosome 4q28.3 and 7q31.2. Haplotype analysis was performed based on haplotype blocks to figure out if there were haplotypes that would be associated with the strabismus phenotype.

Linkage analysis in large pedigrees

Linkage analysis estimates recombination fractions between a putative disease locus and marker loci (Lathrop et al., 1984), and the results were output as LOD (logarithm of odds) scores (Ott, 1999; Terwilliger & Ott, 1994; Terwilliger & Ott, 2016). The Pseudomarker program (Gertz et al., 2014; Goring & Terwilliger, 2000; Hiekkalinna et al., 2011) estimates allele frequencies by maximum likelihood, separately under linkage and no linkage, which makes the results virtually independent of allele frequencies. Since the Pseudomarker program requires error-free data, we had to remove SNPs as necessary as possible to obtain a pure and error-free dataset (Lathrop et al., 1984). In addition, this program can also take linkage disequilibrium (LD) between a SNP and the disease into account, thus resulting in gain of additional power. Linkage analysis generally requires absence of errors. Mendelian inconsistencies would be allowed in linkage analysis with a suitable choice of penetrance for SNPs, but the procedure is cumbersome and rarely done.

Results

Hardy-Weinberg equilibrium and principal component analysis

After quality control of SNPs and individuals, 19 SNPs with monomorphic or undetected types or at a low call rate were excluded, and one SNP was merged to another in the database of NCBI. Finally, all individuals and 237 SNPs remained. The flow chart of analytical process is shown in Fig. 1.

Figure 1 Flow chart of analytical process.

TDT, transmission disequilibrium test; TDTae, TDT allowing for errors.

Twenty one out of 23 SNPs on chromosome 4q28.3, and 208 out of 214 SNPs on chromosome 7q31.2 were in Hardy-Weinberg equilibrium (P > 0.05, chi-square test). The results of PCA analysis showed no population substructure among families or between affected and unaffected individuals, and also detected no outliers (Fig. 2).

Figure 2 Scatter plot of eigenvectors in principal component analysis (PCA).

The horizontal axis is first principal component and the vertical axis is the second principal component. Each circle represents a subject. Open circles represent unaffected subjects and closed circles represent affected subjects. The same color indicates subjects in the same family. Neither the subjects in the same family nor the affected or unaffected subjects as a whole show clusters. There are no outliers.

TDT and TDTae

A total of 261 Mendelian errors were detected. Errors could be due to faulty genotyping, clerical mistakes, pedigree mismatch (e.g., adopted child), or a new mutation which the program treated as an inherited variant. The dataset with Mendelian errors was applied to TDT directly. Mendelian errors were reduced when some families with large error rates were deleted. After five SNPs (rs4148690, rs3757807, rs3807975, rs3779551, and rs213987, all on chromosome 7) and four families (37, 8, 10, and 43) with the highest error rates were deleted, 186 individuals in 54 families and 232 SNPs with 71 errors remained, which were used as the analysis set for TDTae.

The best results of TDT (P < 0.08) are shown in Table 2. And the TDTae results with significant P values (P < 0.05) are shown in Table 3. Figures 3 and 4 show the results of TDT and TDTae, with haplotype analysis and LD blocks adjusted for families on chromosome 4q28.3 and on chromosome 7q31.2, respectively. In the 4q28.3 locus, all significant SNPs were located in the intron of MGST2. In the 7q31.2 locus, several SNPs with significant P values (P < 0.05) were distributed dispersedly in TES, ST7, WNT2, CAV1, and CFTR. Haplotype analysis in the 4q28.3 locus did not get a positive result. In the 7q31.2 locus, haplotypes with significant P values are located in TES, CAV1, or WNT2.

Table 2 Results of transmission disequilibrium test (TDT).

Chromosome	dbSNP	Chromosomal location	Gene	Minor allele	Major allele	Transmitted alleles (no.)	Untransmitted alleles (no.)	Odds ratio	χ2	P value	
7	rs3807961	115855791	TES intron	T	C	19	8	2.375	4.481	0.03426	
7	rs1000966	115862902	TES intron	A	G	6	19	0.3158	6.76	0.009322	
7	rs3807967	115863441	TES intron	G	A	16	4	4	7.2	0.00729	
7	rs3807977	115881740	TES intron	A	T	14	6	2.333	3.2	0.07364	
7	rs3757730	115882323	TES intron	G	A	11	21	0.5238	3.125	0.0771	
7	rs3807979	115893362	TES intron	G	A	19	8	2.375	4.481	0.03426	
7	rs3030648	115897792	TES3′-UTR	T	A	10	27	0.3704	7.811	0.005193	
7	rs2074025	116550456	CAPZA2 intron	C	T	6	1	6	3.571	0.05878	
7	rs3808193	116868696	ST7 intron	A	G	4	0	Infinity	4	0.0455	
7	rs2024233	116917427	WNT23′-UTR	G	A	10	24	0.4167	5.765	0.01635	
7	rs733154	116918911	WNT2 intron	G	A	5	13	0.3846	3.556	0.05935	
7	rs2896218	116919978	WNT2 intron	G	A	7	17	0.4118	4.167	0.04123	
7	rs3824028	116933900	WNT2 intron	G	A	0	3	0	3	0.08326	
7	rs1049337	116200587	CAV13′-UTR	T	C	4	0	Infinity	4	0.0455	
7	rs2299442	117131336	CFTR intron	A	T	0	4	0	4	0.0455	
4	rs1000222	139666831	MGST2 intron	C	T	8	18	0.4444	3.846	0.04986	
4	rs3836606	139701270	MGST2 intron	DEL	G	2	12	0.1667	7.143	0.007526	
Notes.

P values are nominal and not corrected for testing multiple SNPs.

Table 3 Results of transmission disequilibrium test allowing for errors (TDTae).

Chromosome	dbSNP	Chromosomal location	Gene	aR2	cMinimum of corrected P value	
				Dominant model	Recessive model	Multiplicative model		
				bDSB	GHLO	DSB	GHLO	DSB	GHLO		
7	rs3807959	115853100	TES intron			4.4081	4.4081	12.5929	12.5929	0.008324	
7	rs3807967	115863441	TES intron					0.1522	0.1522	0.0416597	
7	rs3807986	116177825	CAV1 intron				2.0739			0.0471031	
7	rs3801993	116190382	CAV1 intron			8.2562	8.2562			0.0378839	
7	rs3801994	116190469	CAV1 intron			8.3368	8.3368			0.0366007	
7	rs41735	116435416	MET intron			0.13	0.13	0.1164	0.1164	0.019151	
7	rs2023748	116436022	MET intron			0.1482	0.1482	0.1228	0.1228	0.034239	
7	rs41737	116436097	MET			0.2547	0.2547			0.0453541	
7	rs2301649	116538634	CAPZA2 intron			0.3377				0.033912	
7	rs2074025	116550456	CAPZA2 intron					0.0169	0.0169	0.016231	
7	rs38861	116816284	ST7 intron	3.3527	3.3527			6.3843	6.3843	0.018479	
7	rs3735646	116915684		3.3	3.3					0.0439441	
7	rs2024233	116917427	WNT2 3′-UTR	6.8558	6.8558			6.8744	6.8744	0.009829	
7	rs3779547	116930962	WNT2 intron			0.1892	0.1892			0.0491057	
7	rs3779546	116934200	WNT2 intron			0.1779	0.1779			0.031462	
7	rs2285544	116944283	WNT2 intron	3.698	3.698					0.023326	
7	rs4148721	117267954	CFTR intron					19.973	19.973	0.0465618	
4	rs3836607	140579303		0.1473	0.1473			0.0476	0.0476	0.019239	
Notes.

a R1 = R2 in dominant mode of inheritance; R1 = 1 in recessive mode of inheritance; R12  = R2 in multiplicative mode of inheritance.

b Douglas Skol Boehnke (DSB) error model; Gordon Heath Liu Ott (GHLO) error model.

c The minimum P value (corrected for multiple testing) of dominant, multiplicative, and recessive mode of inheritance.

The corrected P value is given by 1 –(1 –p)k−1 (Gordon et al., 2004).

R1 = Pr(aff | + d)/Pr(aff | +  +) and R2 = Pr(aff |dd)/Pr(aff | +  +) are genotypic relative risks for a di-allelic trait locus with low-risk (wild-type) allele + and high-risk (disease) allele d. If both R1 and R2 are less than 1, the genotypic relative risk value of the other allele would be calculated by R1′ = R1/R2 and R2′ = 1/R2. A few strange results are omitted from this table (e.g., R > 10,000, or R = 0).

Figure 3 TDT (transmission disequilibrium test) and TDTae (TDT allowing for errors) plots, and linkage disequilibrium (LD) blocks on the 4q28.3 locus.

(A) Physical position of SNPs in the genome (4:133152903-139701270, release 108), and annotation track is displayed by Integrative Genomics Viewer (IGV) 2.3. (B) The χ2 and −log10(P) value in TDT, and (C) −log10(P) value in TDTae. The regions with significant markers (P < 0.05) are highlighted. (D) LD plot, showing LD patterns among the SNPs in TDT analysis. The LD between the SNPs is measured as D′ value and shown (×100) in the diamond at the intersection of the diagonals from each SNP. D′ < 1 and LOD < 2 is shown as white, D′ = 1 and LOD < 2 is shown as blue, D′ < 1 and LOD ≥ 2 is shown as shades of pink or red, and D′ = 1 and LOD ≥ 2 is shown as bright red. Haplotype blocks in high LD are outlined in bold black line.

Figure 4 TDT (transmission disequilibrium test) and TDTae (TDT allowing for errors) plots, and linkage disequilibrium (LD) blocks on the 7q31.2 locus.

(A) Physical position of SNPs in the genome (7:116210883-117667022, release 108) and annotation track is displayed by Integrative Genomics Viewer (IGV) 2.3. (B) The χ2 and −log10(P) value in TDT, and (C) −log10(P) value in TDTae. The regions with significant markers (P < 0.05) are highlighted. (D) The haplotype block structure from case-parent trios. There are nine haplotype blocks at the locus. The SNP numbers on the top of haplotypes correspond to those in the diagram of ped files. The haplotype frequencies are shown on the right of each haplotype. (E) LD plot, showing LD patterns among the SNPs in TDT analysis. The LD between the SNPs is measured as D′ value and shown (×100) in the diamond at the intersection of the diagonals from each SNP. D′ < 1 and LOD < 2 is shown as white, D′ = 1 and LOD < 2 is shown as blue, D′ < 1 and LOD ≥ 2 is shown as shades of pink or red, and D′ = 1 and LOD ≥ 2 is shown as bright red. Haplotype blocks in high LD are outlined in bold black line.

Linkage analysis

Four families and 46 SNPs with Mendelian errors were deleted. Furthermore, linkage analysis requires valid data of complete family trios, so the program extracted only 90 individuals in 21 families which remained in the final analysis set. The dominant mode was less plausible because few parents in the present families were affected in themselves, or rather, the trait under consideration was frequently consistent with recessive inheritance. Table 4 shows all significant SNPs under dominant inheritance and recessive inheritance. The linkage analysis employed the error-free dataset which were only available by reducing the number of families, individuals, and SNPs. Based on this methodological limitation of linkage analysis, the LOD scores in the present analysis with the small number of families were as small as just over 1.

Table 4 Results of linkage analysis.

Chromosome	dbSNP	Chromosomal location	Gene	LOD score	Linkage P value	Model	
7	rs3757729	115861846	TES intron	1.263692	0.007935	Recessive	
7	rs1004109	115862296	TES intron	1.051778	0.013885	Recessive	
7	rs3807967	115863441	TES intron	1.051121	0.013909	Dominant	
7	rs3757730	115882323	TES intron	1.283524	0.007534	Recessive	
7	rs3823977	115893407	TES intron	1.051778	0.013885	Recessive	
7	rs3807983	115898991	TES3′-near	1.051777	0.013885	Recessive	
7	rs3840660	116917245	WNT23′-UTR	1.43437	0.005093	Recessive	
7	rs3779550	116927360	WNT2 intron	1.064359	0.013427	Recessive	
7	rs213976	117238379	CFTR intron	1.129688	0.011289	Recessive	
7	rs213977	117238445	CFTR intron	1.720078	0.00245	Recessive	
7	rs4148714	117238453	CFTR intron	1.725551	0.002416	Recessive	
7	rs2246450	117240668	CFTR intron	1.505123	0.004244	Recessive	
7	rs2299445	117246315	CFTR intron	1.424525	0.005224	Recessive	
7	rs2237726	117256374	CFTR intron	1.725551	0.002416	Recessive	
7	rs2254742	117264126	CFTR intron	1.721261	0.002442	Recessive	
7	rs214167	117286524	CFTR intron	1.204099	0.009277	Recessive	
7	rs4148724	117305151	CFTR intron	1.806151	0.001969	Recessive	
Notes.

LOD logarithm of odds

P values are nominal and not corrected for testing multiple SNPs.

Expression quantitative trait locus (eQTL)

The significant SNPs in the 4q28.3 locus were related to MGST2 transcription in the search for expression quantitative trait locus (eQTL) in the Human Genetic Variation Database (HGVD) which displays the Japanese genetic variations and the association between the variations and transcription levels of genes (Higasa et al., 2016).

Discussion

In our preceding study, we tried to use a method that did not depend on kinship, such as association study adjusted by family, in order to examine the relationship between several polymorphic markers and the strabismus phenotype in the chromosomal regions (Matsuo, 2015). This strategy was based on the fact that some of the family trios were not complete and that there were many Mendelian errors which might be attributed to adoption. The preceding results showed that significant SNPs were in MGST2 and WNT2 on chromosomal 4q28.3 locus and 7q31.2 locus, respectively. However, the false discovery rate (FDR) was too high to reduce the power in conducting multiple comparisons among SNPs (Benjamini & Hochberg, 1995), and therefore, we turned in the present study to focus on methods for linkage analysis.

When a few families or SNPs seem to contribute to the majority of errors, it is best to delete these families or SNPs firstly and then to carry out TDTae. In contrast, TDT by Plink would handle errors by ignoring the offending genotypes. The TDTae program generally furnishes much smaller P values than the TDT since the error model is in a parametric manner. There is some agreement between the outputs from Plink and TDTae, although not very strong. In the present analyses, the two error models in the TDTae furnished similar results, suggesting that the results would not be unduly dependent on the assumptions regarding errors.

The P values shown in the linkage studies were nominal and not corrected for the testing of multiple SNPs. A correction for multiple comparisons was somewhat difficult to make since these SNPs are presumably highly correlated with each other. The Pseudomarker program requires strict error-free data with no Mendelian errors. Therefore, some potential candidates of SNPs might be deleted in the process of preparing error-free data which were based merely on a single faulty genotyping. Furthermore, it is worth noting that genotyping errors would cause inflation in the recombination fraction between the disease and marker loci, leading to the consequence that recombination fractions may appear larger than they truly are Lincoln & Lander (1992).

In the present study, we clearly demonstrated that MGST2 is a candidate for the chromosomal 4q28.3 locus. As for the 7q31.2 locus, in contrast, the results of different kinds of statistical analyses could not narrow the locus to a single gene. Under the circumstances, the distribution of significant SNPs in the locus showed that only the ST7 to WNT2 region contained significant SNPs for all three methods of linkage analysis (Fig. 5). In the 7q31.2 locus, ST7 is indeed in the same big haplotype block with WNT2.

Figure 5 Summary of SNP −log10(P) values for each method of linkage analysis on chromosome 7q31.2.

(A) Physical position of SNPs in the genome. (B) The plots represent −log10(P) values of sigificant P values obtained from different analyses.

Primary and non-syndromic comitant strabismus contains several different clinical entities: esotropia includes infantile esotropia, accommodative esotropia, partially accommodative esotropia, late-onset (acute-onset) esotropia, and microtropia (microesotropia) while exotropia includes intermittent exotropia, constant exotropia and congenital (infantile) exotropia (Matsuo et al., 2003; Matsuo et al., 2005; Matsuo & Matsuo, 2005; Matsuo & Matsuo, 2007; Matsuo et al., 2007). Furthermore, patients with the same clinical entity or clinical diagnosis show varying degrees of manifestations, not only in horizontal and vertical deviations but also in the state of binocular vision. Under the circumstances, one way to define comitant strabismus is as a disease with abnormal binocular vision, namely abnormalities in simultaneous perception, fusion and stereopsis.

In our ongoing research, different clinical entities of primary and non-syndromic comitant strabismus were analyzed altogether in chromosomal mapping and SNP typing (Fujiwara et al., 2003; Shaaban et al., 2009a; Shaaban et al., 2009b; Matsuo, 2015). In other words, the presence or the absence of a phenotype “strabismus” was used as a single phenotypic descriptor in the genetic statistical analysis. This approach was justified by the fact that in our previous study the same chromosomal susceptibility loci were replicated in stratified groups of the families either with esotropia or with exotropia (Shaaban et al., 2009a).

In the Japanese population, exotropia is more prevalent than esotropia (Matsuo & Matsuo, 2005; Matsuo & Matsuo, 2007; Matsuo et al., 2009b; Matsuo et al., 2010), in contrast with the Caucasian population which shows higher prevalence of esotropia. A common genetic mechanism is assumed to give rise to exotropia and esotropia since both entities of comitant strabismus share abnormal binocular vision as a phenotype. In addition, there are indeed families which show mixed phenotypes of exotropia and esotropia, as observed in this study: one member shows exotropia and another member shows esotropia in a family. Abnormal activities of unknown genes in the central nervous system might be responsible for the abnormal binocular vision in patients with comitant strabismus.

The large number of Mendelian errors seems to be the limitation of this study. The original data sets with Mendelian errors were used in TDT analyses (Spielman & Ewens, 1996; Gordon et al., 2004). In contrast, the error-free data sets were used in TDTae and linkage analyses under dominant and recessive inheritance (Lathrop et al., 1984). The common use of the original data sets should have underlain the more consistent results whereas sharing of the same data sets would not necessarily mean that the data applied actually in analysis are the same. Or rather, the difference is merely the methods to handle the errors prior to software application or in the process by ignoring a single cell or deleting the whole series. In the present study, permutation tests were done to check the robustness of the results. Therefore, the results should be affected mainly by the methods, and would not be decided by the number of Mendelian errors.

Both MGST2 and WNT2 are known to be expressed in the brain (Jakobsson, Mancini & Ford-Hutchinson, 1996; Cadigan & Nusse, 1997) and likely to be involved in the development of comitant strabismus. Different analytical methods shed light on the data from different angles, therefore it is useful to apply more than one type of analysis. Strict Mendelian application as in this study, might not be appropriate in multifactorial disorders such as comitant strabismus, but would certainly provide a step to get guidance for detecting genetic risks of the disease. Since the proof of a responsible gene in a multifactorial disorder is difficult to be obtained in animal experiments, a different approach in patients, such as whole exome sequencing, would provide support for the present results of SNP typing. Further functional studies are necessary to clarify the mechanisms of the two genes on the susceptibility of comitant strabismus. Finally, it should be noted that there is a limitation in applying the eQTL to the present study since the eQTL data have been obtained in analyses of blood cells (Higasa et al., 2016).

Conclusions

This study with different analytical methods for genetic statistics provides evidence that MGST2 and WNT2 are potential candidate genes for comitant strabismus in Japanese population.

Supplemental Information

Data S1 Raw data

The terms are sample ID, groups and varient call information and pedigree information.

Click here for additional data file.

Linkage analyses in this study were done in collaboration with Dr. Jurg Ott at Rockefeller University, New York, NY, USA, and Dr. Atsuko Imai at Departments of Cardiovascular Medicine and Genome Informatics, Osaka University Graduate School of Medicine, Osaka, Japan. We thank Dr. Kazuhiro Sato at Institute of Plant Science and Resources, Okayama University for his valuable discussion. We also thank Yangyang Liu, Department of Epidemiology, Okayama University Medical School and Graduate School of Medicine, Dentistry, and Pharmaceutical Sciences for his advice on statistics.

Additional Information and Declarations

Competing Interests

Author Contributions

Human Ethics

Data Availability

The authors declare there are no competing interests.

Jingjing Zhang performed the experiments, analyzed the data, wrote the paper, prepared figures and/or tables, reviewed drafts of the paper.

Toshihiko Matsuo conceived and designed the experiments, performed the experiments, analyzed the data, contributed reagents/materials/analysis tools, wrote the paper, prepared figures and/or tables, reviewed drafts of the paper.

The following information was supplied relating to ethical approvals (i.e., approving body and any reference numbers):

The study followed the tenets of the Declaration of Helsinki, and was approved by the Ethics Committee of Okayama University Graduate School of Medicine, Dentistry, and Pharmaceutical Sciences. Approval number: Genome #215.

The following information was supplied regarding data availability:

The raw data has been provided as Data S1.

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
