# Peer review of "MGST2 and WNT2 are candidate genes for comitant strabismus susceptibility in Japanese patients"

_PeerJ, doi:10.7717/peerj.3935_

## Round 0.1 · original submission · Major Revisions

The statistical analysis in this paper has a serious problems, as described by both of the referees. Other than the association analysis that treats the subjects as unrelated,the components are not erroneous per se but there the analysis is flawed in the way the components fit together. I strongly recommend that authors add an additional author, namely a statistical geneticist.

The reviewers have provided detailed notes both about the shortcomings in the manuscript and the ways that it could be improved. I endorse all of these and urge the authors to pay very close attention to the reviews when revising the manuscript.

·

Basic reporting

All information contained in attached review.

Experimental design

All information contained in attached review.

Validity of the findings

All information contained in attached review.

Reviewer 2 ·

Basic reporting

General comments:
The language in this manuscript needs to be improved to a more professional and less conversational style. A review by a native English speaker should be sufficient to raise the manuscript up to an appropriate standard and address the overuse of commas throughout the paper.

The manuscript seems well referenced and the background in the main body of the paper describes the phenotype well but should expand slightly on the previous work by these authors.

The manuscript structure is somewhat inconsistent. The introduction section needs only some language refinement but the method section is very confusing and hard to follow the sequence of work. As a result, interpreting the results and conclusions is impossible. The tables are poorly formatted and contain confusing information which is not always relevant and appropriate. The raw data was available to this reviewer.

The manuscript appears self-contained, has clear aims and a relevant result reported which is consistent with the hypothesis.

Specific Comments:
Abstract: The Section Background/aim should have a more detailed description of the phenotype since this is a general interest journal, not a ophthalmology journal. A sentence or two should suffice to explain the features to the reader.

Introduction: Lines 73-80. The language here needs to be improved to a more professional and less conversational style.

Figures: The LD figures (1 and 2) are incorrect since no adjustment for kinship has been made and the association analysis they support is entirely inappropriate.

Tables: Table 1 should be completely revised. Table 2 contains a column labeled "Other Methods" which is hard to understand. Table 3 contains an eQTL column which adds nothing to the results.

Experimental design

General Comments:

This is original primary research well within the aims and scope of the journal.

The research question of whether SNPs in two previously identified locations by prior work, is well defined, relevant and meaningful. This is a classic fine-mapping approach which seeks to refine a prior signal in two genomic locations and the family-based study design is appropriate in this context.

The rigor of the investigation was harder to assess. The methods section is confused and poorly structured and raises several questions in this reviewers mind as to the statistical understanding of the authors. They make several assertions that are incorrect, misleading or outdated that significantly undermine their results. Replicating their analyses with the methods section in its current form would not only be quite difficult but would in fact be inappropriate in some parts. I address these failings in the specific comments below.

Specific Comments:

The methods section need major revision in order to meet even basic standards of adequate reporting. The way it is currently structured suggests that certain quality control measures were applied after some analyses were performed. Since this seems unlikely, I suggest the authors use a flowchart to clarify the workflow and use this to restructure the methods section. This flowchart could be added to the paper if figure limits allow or added to a supplementary methods section.

For the description of the study subjects, the table is poorly formatted and should be revised to reduce redundancy. It is typical to include pedigree drawings unless there is a risk of identification and would greatly assist the reader in understanding the cohort. Samples with available DNA could be indicated on these drawings, rather than detailed in the table.

The section describing the genotyping seems straightforward but a table of the primers should be included in the supplementary methods. However, the genotyping rate is rather lower than I would expect and does raise some concerns which the authors should address. Why were there such high missingness rates?

This manuscript contains no discussion of power. Power calculations should be performed and included in the methods and results.

This manuscript contains no discussion of population stratification. Even within apparently homogeneous populations, significant substructure can still exist and can influence the results of association based tests. A discussion of this in the methods would be appropriate. Although linkage analysis is robust to population stratification, it does require good estimation of allele frequencies in the data. This should be included in the methods.

I also have several concerns about how the analyses were performed. The section on association testing uses a completely inappropriate methodology to analyze these data. The authors even seem aware of this themselves, stating that standard chi-square analyses assume independence of the subjects and this standard is not met in this cohort. There is no defending this analysis, it is meaningless and should be removed. A more appropriate association approach would be to use FBAT, EMMAX (with the appropriate caveats about the beta generated) or other methods designed to use or control for kinship between study participants.

The next section discusses Mendelian inconsistencies and this section raises several red flags. The authors mention duplicate subjects but do not adequately explain how duplicate subjects were selected for removal. Equally of concern is the large number of reported Mendelian errors and the methods do not sufficiently detail how these errors were dealt with. Typically, one would remove inconsistent genotypes in a SNP just in a family if only one family had an error. If the same SNP has errors in two or more families, the SNP should be deleted across the entire dataset. It is not clear whether the authors did this in their data.

Although the authors discuss using TDTae, a method to address genotyping errors without deleting data, this is not used widely in the field as it has been superseded by other, better tests. In addition, the authors describe this method as parametric, which is not exactly correct - the error model is parametric but the test for linkage is not.

But no discussion of the other analyses described in the results is included here. These sections should be revised and reordered.

Validity of the findings

The association results are the result of an inappropriate methodology and are meaningless. This section should be removed or the analysis repeated with a more appropriate test as I suggested in my review of the methods section. The results also contain considerable discussion of the methods and these portions should be moved to the methods section. For example, there is no mention of the pseudomarker program until the reader reaches the results section. This is not the right way to report methods or results. The exact models used should be presented and the method used for generating allele frequencies.

As I've noted, methods are poorly described and as a result, it is hard to assess whether any of the linkage results presented here are meaningful, as I cannot ascertain whether the data were appropriately cleaned before analysis. The use of TDTae is discouraged and it is not clear to me if the authors fully understand the implications of using this test in these data.

Finally, the authors make no adjustment for multiple comparisons in these data, despite using several analysis methods and in the parametric linkage analysis, more than one model. It seems likely that such an adjustment would render all results entirely non-significant but should be performed and the manuscript adjusted to reflect the results in the light of this.

Additional comments

Although this manuscript presents an interesting fine-mapping study, I have grave concerns about the statistical analyses presented and as a result, cannot recommend this paper for publication without significant revision.

---

## Round 0.2 · Minor Revisions

I appreciate the revisions that you have made. However, one problem still remains and one problem has been introduced.

I find the justification for using an analysis as if the design were case-control to be entirely unconvincing. You will have to remove this approach entirely if the paper is to appear in PeerJ.

The use of Fisher's method of combining p-values is unfortunately inappropriate. Fisher's method is for *independent* tests. As your tests are on the same sample, they are not independent.

I appreciate that your data has the unusual feature of somewhat degraded DNA, with the consequence of more genotyping errors than usual. Your use of the TDTae is a reasonable attempt to handle that (while the case-control analysis is not).

I have marked the decision as "minor revisions" because if you remove the material on case-control analysis and on Fisher's method I will be able to make a final decision without sending the paper out again to reviewers.

·

Basic reporting

No comment.

Experimental design

No comment.

Validity of the findings

No comment.

---

## Round 0.3 · accepted · Accept

Thank you for your responsive changes in the manuscript.